# Comparative Morphology of the Carnassial Teeth Root Canals in Mixed-Breed Dogs and German Shepherds

**DOI:** 10.3390/ani14081138

**Published:** 2024-04-09

**Authors:** Faruk Tandir, Rizah Avdić, Nejra Dučić, Aida Džanković, Redžep Tandir, Ermin Šaljić, Anel Vejzović, Nedžad Hadžiomerović

**Affiliations:** 1Department of Basic Sciences of Veterinary Medicine, Veterinary Faculty, University of Sarajevo, 71000 Sarajevo, Bosnia and Herzegovina; rizah.avdic@vfs.unsa.ba (R.A.); nejra.ducic@vfs.unsa.ba (N.D.); anel.vejzovic@vfs.unsa.ba (A.V.); nedzad.hadziomerovic@vfs.unsa.ba (N.H.); 2Department of Dental Pathology with Endodontics, Faculty of Dentistry with Dental Clinical Centre, University of Sarajevo, 71000 Sarajevo, Bosnia and Herzegovina; aidadzankovic@gmail.com; 3The Public Institution Health Centre of Sarajevo Canton—Dental Service, 71210 Sarajevo, Bosnia and Herzegovina; rtandir@yahoo.com; 4Department of Clinical Sciences of Veterinary Medicine, Veterinary Faculty, University of Sarajevo, 71000 Sarajevo, Bosnia and Herzegovina; ermin.saljic@vfs.unsa.ba

**Keywords:** endodontic treatment, dog, root, apical delta, carnassial teeth

## Abstract

**Simple Summary:**

The aim of this study was to analyze and compare the morphology of root canals in the carnassial teeth of German shepherds and mixed-breed dogs. Knowledge of the morphology of root canals is essential for a successful endodontic procedure. It has been determined that an apical delta, which represents a complex structure that consists of multiple cavities whose purpose is to allow the passing of nerves and blood vessels from the pulp cavity to the apex of the root, was present in 247 roots. There are four main types of apical delta. The most common type in superior fourth premolars was type II, with up to 10 apical ramifications, whereas type IIIA, with 10–20 apical ramifications, was most commonly present in inferior first molars.

**Abstract:**

Root canal treatment of carnassial teeth in dogs is a common endodontic technique which aims to re-establish or maintain the health of the periapical tissues. In total, 43 dogs were used in this study. Root canal morphology was evaluated in 86 superior fourth premolars and 86 inferior first molars. Apical delta was present in 247 roots, while obliteration of the root canal was found in 11 roots. The most common type of apical delta of the roots of superior fourth premolars was type II, with up to 10 apical ramifications, while type IIIA, with 10–20 apical ramifications, was most commonly present in the roots of inferior first molars. Considering that knowledge of the morphology of root canals is essential for a successful endodontic procedure, the aim of this study was to analyze and compare the morphology of root canals in the carnassial teeth of German shepherds and mixed-breed dogs. Apical resection for the purpose of endodontic therapy of the superior fourth premolar and the inferior first molar is indicated at a length of 4 to 6 mm from the anatomical tip of the roots, which would completely remove the apical delta of these two teeth.

## 1. Introduction

The superior fourth premolar and the inferior first molar, also known as the carnassial teeth, and the canine teeth are considered to be the most important teeth in a dog regarding their proportions, form, and purpose [1]. The expression “carnassial” signifies flesh cutting, and these teeth are thought to be the largest shearing teeth in the mouth [2]. The carnassial teeth are morphologically quite complex, which can be correlated with their distinct functions [3]. Various pathological disorders, including fractures, can affect these teeth, and most of these disorders require extraction or endodontic treatment [4]. Root canal treatment is a common endodontic technique, the aim of which is to re-establish or maintain the health of the periapical tissues as an alternative to tooth extraction [5]. The success of endodontic treatment depends, to a large extent, on detailed cleaning, shaping, filling, and obturation of root canals, which requires in-depth knowledge of the tooth anatomy and root canal morphology. The most common cause of failure of endodontic treatment is incomplete filling and closure of the canal [6,7,8]. The apical delta and the adjacent tissues are of great importance in the process of root canal obturation [9]. Several studies [10,11,12] have shown that the main root canal in dogs ends in an apical delta. According to Watanabe et al. [13], the apical delta represents a complex structure which consists of multiple cavities whose purpose is to allow the passing of nerves and blood vessels from the pulp cavity to the apex of the root. The root in immature dogs has an incomplete, open apex. Maturation of teeth results in the formation of an apical delta at around eighteen months of age in dogs, regardless of the type of teeth. Usually, one main pulp canal for each root is present along with multiple collateral canals which flow parallel to the main one [1].

One of the techniques used in visualization of the morphology of root canals is the clearing technique, which implies demineralization and tooth staining. Clearing is achieved in three stages: decalcification, dehydration, and clearing itself. This procedure is performed for the purpose of research in a laboratory, and it cannot be used in clinical practice. The clearing technique is considered a simple and excellent method that provides a three-dimensional evaluation of the morphology of root canals [14,15]. Considering that understanding of root canal anatomy is essential for the success of endodontic treatment, the aim of this study is to provide a three-dimensional view of the root canal system and investigate, using the clearing technique, the shape of the root canals of the superior fourth premolar and the inferior first molar in mixed-breed dogs and German shepherds.

## 2. Materials and Methods

In total, 43 dogs were included in the study; of these, 35 were mixed-breed dogs and 8 were German shepherds. The average age of German shepherds was 5.5 years, whereas the average age of mixed-breed dogs was 2.2 years. The research included 172 teeth: 86 superior fourth premolars (258 roots) and 86 inferior first molars (172 roots), i.e., a total of 430 roots. The study materials were obtained from euthanized animals between 2015 and 2021. The animals were treated in accordance with the relevant legislation [16]. The length of the roots was measured from the cemento-enamel junction to their apex.

The access cavity was made on the occlusal surface of the crown; this provided unrestricted access to all root canals. Each tooth was immersed in a 10% formalin solution and individually stored in a plastic container which was numbered and indicated the breed, age, and sex of the dog. Calculus and pigmentation from the teeth were removed by immersion in an ultrasonic bath for a duration of 20 min. Residual dental deposits were removed by immersing the samples in a 5.25% solution of sodium hypochlorite for 2 h. In order for the pulp chamber to be accessible, an access cavity was made using a dental drill with round and conical diamond burs with a diameter of 1.6–2.3 mm. After accessing the cavity, root canal cleaning needles of different diameters were inserted into the pulp chamber in order to remove the remains of the pulp tissue. The second stage of decomposition of organic tissue in the root canals took 12 h by immersing the teeth in a 5.25% sodium hypochlorite solution followed by washing the samples under running water for 2 h. When the root canal was completely dry, a few drops of Indian ink were injected at the root canal entrances using a 26 G needle. After drying, the openings of the root canals were closed with zinc oxide sulfate cement, and the rest of the access cavity was filled with pink dental wax. Samples prepared in this way were immersed in 5% nitric acid at room temperature for 13 days, with daily changes of the acid (every 24 h). As a result of decalcification, the teeth became flexible and elastic, and it became possible to bend the roots. The degree of decalcification was measured by inserting a dental probe into the crown of the tooth. The smooth passage of the probe through the crown of the tooth indicated that the end of the decalcification process was reached. After demineralization, the samples were rinsed with running water for 4 h and then gradually placed in increasing concentrations of ethyl alcohol (70%, 96%, and 99.5%) for 12 h for each concentration. By immersing the teeth in ethyl alcohol, complete dehydration of the teeth is achieved, which leads to the loss of elasticity of the teeth. After drying the samples, they were immersed in methyl salicylate, which achieved transparency of the teeth. The number of root canals within an individual root, the presence of the obliteration of root canals, the presence and position of lateral canals, and the presence and type of apical deltas were observed using an optical magnifier with 10× magnification. In addition, the width of the root at the point where the apical delta began, as well as the length of the apical delta, from the beginning of the apical ramification to the anatomical tip of the tooth, were determined with an optical magnifier (Figure 1).

## 3. Results

The research results show the complex anatomical structure of the roots of the superior fourth premolar and inferior first molar. The average lengths of these teeth are shown in Table 1.

By analyzing the root system in 258 roots of superior fourth premolars, it was observed that a passable root canal ending with apical openings was present in 247 roots, while obliteration of the root canal was found in 11 roots in the upper and middle third of the roots, all of which belonged to mixed-breed dogs. In total, 4.26% of obliterated canals were observed, of which 1.1% were present in the mesiobuccal roots, 2.7% in the mesiopalatal roots, and 0.38% in the distal roots (Figure 2).

The occurrence of ramification of the main root canal into smaller canals in the apex area leads to the formation of a characteristic apical termination in the form of an apical delta. Almost all the analyzed roots of superior fourth premolars had an apical delta, except for the roots which were obliterated. On the basis of the ramification of the root canals, the apical delta was classified into four types, modified according to Watanabe et al. [13]:Type I represents a root canal with one wider apical opening, without the apical delta. This type was not recorded in either the superior fourth premolars or inferior first molars.Type II: a few apical deltas where the root canal ends with up to 10 ramifications. This type was present in 185 roots of the superior fourth premolar and 71 roots of the inferior first molar (Figure 3 and Figure 4).Type IIIA: a low apical delta where the root canal ends with 10 to 20 ramifications. In our research, this type was recorded in 60 roots of the superior fourth premolar and 87 roots of the inferior first molar (Figure 5 and Figure 6).Type IIIB: a high apical delta, where the root canal ends with over 20 ramifications, which was noted in two roots of the superior fourth premolar and one root of the inferior first molar (Figure 7 and Figure 8).

The root width at the point of initiation of apical ramification in the root canals of superior fourth premolars and inferior first molars was measured, and the results are shown in Table 2.

In mixed-breed dogs, differences were observed between the mesiobuccal and distal, as well as between the mesiopalatal and distal roots of the right superior fourth premolars (tooth 108). Such differences were not noted between the mesiobuccal and mesiopalatal roots of this tooth. Identical differences were observed on the roots of the left superior fourth premolars (tooth 208). The width of the roots at the point of apical ramification in the root canals of tooth 108 were similar to the distal roots, which ranged from 6.06 to 1.73 mm. The mesiopalatal roots had a width of 1.05 to 4.29 mm, while the mesiobuccal roots were the narrowest (0.53–3.84 mm). In tooth 208, these values were also the most noticeable in the distal roots and ranged from 1.71 to 7.64 mm. The lowest values were observed in the mesiopalatal roots (0.86–4.83 mm), while for the mesiobuccal roots, the results ranged from 1.30 to 5.31 mm. Numerical values in German shepherds were quite uniform. In tooth 108, the mesiobuccal roots had the lowest values, with a range of 1.81 to 4.49 mm, while the distal roots had the highest values (2.31–6.19 mm).

The root canal of inferior first molars was investigated in 86 samples: 43 left (tooth 309) and 43 right (tooth 409) inferior first molars. Given that the inferior first molars are two-rooted teeth, a total of 172 roots were analyzed, of which 86 were mesial and 86 were distal roots. No obliteration of the root canal was observed in any of these roots. The root widths of inferior first left molars (tooth 309) in mixed-breed dogs were notably different. These differences existed between the mesial and distal roots of this tooth, as well as between the distal roots of tooth 309 and tooth 409. Root widths of tooth 309 ranged from 1.01 to 7.70 mm for mesial roots and from 1.88 to 7.06 mm for distal roots. The widths for the mesial roots of tooth 409 ranged from 1.21 to 8.40 mm, while for distal roots, they ranged from 2.23 to 7.68 mm. In German shepherds, there were no noteworthy differences in this group. The widths of tooth roots in tooth 309 ranged from 2.11 to 7.53 mm in the mesial roots, while in distal roots, they ranged from 3.01 to 6.64 mm. The results were similar in tooth 409, where the values for mesial roots were 2.54–9.50 mm and, for distal roots, 3.73–7.29 mm.

Analysis of the length of the apical delta from the site of initiation of apical ramification to the anatomical apex of the roots of the superior fourth premolars and inferior first molars was also performed. The results are shown in Table 3. In mixed-breed dogs, the only noteworthy difference was observed between the distal roots of tooth 108 and tooth 208. The lengths of the apical delta at the roots of tooth 108 ranged from 0.57 to 3.84 mm in mesiobuccal roots, from 0.38 to 4.41 mm in mesiopalatal roots, and from 0.47 to 4.50 mm in distal roots. In tooth 208, the longest apical delta was observed in distal roots (0.34–7.36 mm). The mesiopalatal roots had the lowest values (0.10–4.65 mm), while the mesiobuccal roots ranged from 0.22 to 4.75 mm. The lengths of the apical delta of the roots in teeth 108 and 208 in German shepherds were not notably different among the roots of the same tooth or between tooth 108 and tooth 208.

In mixed-breed dogs, notable differences were observed between the mesial and distal roots in tooth 309. Such differences were also observed between the distal roots of tooth 309 and tooth 409. The apical delta lengths of tooth 309 ranged from 0.50 to 4.41 mm in the mesial roots, while in the distal roots, they ranged from 0.31 to 3.67 mm. In relation to tooth 409, those values ranged from 0.16 to 5.28 mm for the mesial roots, and from 1.11 to 4.68 mm for the distal roots. The lengths of the apical deltas of teeth 309 in German shepherds ranged from 1.72 to 4.75 mm in the mesial roots, while in the distal roots, they ranged from 1.51 to 4.68 mm. In tooth 409, the mesial roots had values from 2.04 to 5.49 mm, and the distal ones measured from 1.77 to 4 mm.

## 4. Discussion

The presence of the apical delta as a complex cavity system, which enables the unhindered passage of blood vessels and nerves from the pulp cavity to the apex of the roots, has also been confirmed by other authors [13,17,18,19,20,21,22]. Hess et al. [23], in their research of human teeth, concluded that teeth roots do not end with a single apical opening but with an apical delta. Our results coincide with the results of previous authors. Namely, almost all the analyzed roots of the superior fourth premolars and inferior first molars had an apical delta. The exception was the 4.26% of obliterated roots of the superior fourth premolars. This phenomenon was not seen in inferior first molars. Hernández et al. [1] conducted their research on 72 superior fourth premolars and 59 inferior first molars of dogs and also concluded that the main canal ends with an apical delta in each root, in accordance with our results, as well as those of Masson et al. [10] and Gioso et al. [24].

Watanabe et al. [13] classified the apical delta into four types. This classification was used in this research. Considering that all the dogs examined were older than one year, the type I apical delta did not exist either in the superior fourth premolars or in the inferior first molars. A type II apical delta was recorded in 60 roots of superior fourth premolars and 101 roots of inferior first molars. A type III apical delta was recorded in 185 roots of superior fourth premolars and 82 roots of inferior first molars, whereas a type IV apical delta was recorded in 2 roots of superior fourth premolars and 16 roots of inferior first molars. Watanabe et al. [13], in their research, examined the types of apical deltas of permanent teeth of 33 dogs (208 teeth, 314 roots) aged from 6 to 24 months. In six-month-old dogs, they found type I apical deltas in 53% of 34 roots. Types III and IV apical delta were observed in 38% of all roots. In seven-month-old dogs, a type IV apical delta was observed in 76%, while types I, II, and III were very rare and were observed only in canines and premolars. Eight-month-old and nine-month-old dogs all had type IV apical deltas. In adult dogs, the results were different. Types I and II were not observed at all, while type IV was most commonly present. Type III was found in 15 roots (5%) of teeth from 10 dogs, except for canines. Dogs with this type were aged from 5 to 13 years. No type I apical delta was observed in our research, which can be justified by the fact that our research did not include dogs younger than one year old. Our research shows that the presence of type III was the most common, which does not correspond with the results of Watanabe et al. [13], who found type IV to be most abundant. Since their research included all teeth and not just the superior fourth premolars and inferior first molars, one might assume that this is one of the reasons for the disparity between our results and those of Watanabe et al. [13].

The length of the apical delta was measured from the beginning of the apical ramification to the anatomical root tip. In mixed-breed dogs, the length of the apical delta in the roots of superior fourth premolars was 2.07 mm (±0.87 mm) in mesiobuccal roots (*n* = 69). The mesiopalatal roots (*n* = 65) had an apical delta length of 2.29 mm (±0.84 mm), while the lengths of the apical delta of the distal roots (*n* = 71) were 2.71 mm (±1.09 mm). Similar results were obtained by Hernandez et al. [1], where the lengths of the apical delta in 216 roots of the superior fourth premolars had values of less than 1 mm in two roots (0.9%). In 211 roots (97.7%), the length ranged from 1 to 3 mm, while three roots (1.4%) had a length greater than 3 mm. Only five roots in the research by Hernandez et al. [1] differed from our results; that is, they had longer or shorter apical deltas, from 1 to 3 mm. By comparing the results of the lengths of the apical delta on the roots of the superior fourth premolars of German shepherds, it was noted that they had a slightly longer apical delta compared to mixed-breed dogs. In the case of mesiobuccal roots (*n* = 14), the value was 2.48 mm (±0.93 mm). The length of the apical delta of the mesiopalatal roots (*n* = 14) was 2.43 mm (±1.18 mm), while that of the distal roots (*n* = 14) was 3.03 mm (±1.03 mm). These measurements were similar to those determined by Hernandez et al. [1]. The lengths of the apical delta of the inferior first molars from the beginning of the apical ramification to the anatomical tip of the root were also determined by breeds and genders. The apical delta of roots of these teeth in mixed-breed dogs were approximately the same length as the roots of the superior fourth premolars. The mean length of the apical delta of the mesial roots (*n* = 72) was 2.67 mm (±1.01 mm). In the distal roots (*n* = 72), the length was 2.33 mm (±0.78 mm). There were no significant differences between these two roots in mixed-breed dogs. The results of Hernandez et al. [1] were obtained from 59 inferior first molars, that is, 118 roots, and showed that 2 roots (1.2%) were less than 1 mm in length. In 95 roots (80.5%), the length ranged from 1 to 3 mm, and in 21 roots (17.8%), the apical delta was longer than 3 mm. The length of the apical deltas in the mesial roots in German shepherds in our research (*n* = 14) was 3.31 mm (±1.27 mm). The mean value of the length of the apical delta of the distal roots was 2.87 mm (±0.99 mm). The results of Hernandez et al. [1] are very similar to our results, with values of 1–3 mm and >3 mm. However, values of <1 mm, which we did not record, do not match our results. Hernandez et al. [1] pointed out that in their results, 12–18% of the total root length represents the apical delta. Gamm et al. [11], in their study, state that the length of the apical delta in the root canals of canine teeth ranges from 1 to 3 mm and, as such, was present in 52.7% of the analyzed roots. Root widths at the point of initiation of the apical ramification in root canals in mixed-breed dogs were notably greater in both roots of the inferior first molars in comparison to the roots of superior fourth premolars. In the case of inferior first molars, the root widths did not differ from each other, while in superior fourth premolars, the width of the distal root was notably greater compared to the mesiobuccal and mesiopalatal roots.

## 5. Conclusions

Given that there are multiple apical openings in dog teeth, it is impossible to perform mechanical preparation of the apical delta, which is why it is it is necessary to perform endodontic surgery. Research into the morphology of the root canals did not establish the existence of two or more root canals within one root in the superior fourth premolar and the inferior first molar. The analyzed roots with an apical delta in dogs older than one year had multiple apical openings (foramina apicis). The most common apical deltas of the roots of superior fourth premolars have up to 10 apical ramifications, while 10–20 apical ramifications are most commonly present in the roots of inferior first molars. An in vitro study like this one is the most effective strategy to alert veterinary dentistry therapists to the existence of apical deltas. Due to the prevalence of apical delta, endodontic therapy alone is insufficient, and endodontic surgery must be considered as part of the treatment. Radiography of the teeth clearly defines the presence and number of root canals; however, it is very insufficient for interpretation of the apical ramifications and the presence of lateral canals in any part of the root. Therefore, this diagnostic technique should be carried out with caution for endodontics purposes given the impossibility of visualizing the apical delta, as well as the lateral canals. Apical resection for the purpose of endodontic therapy of the superior fourth premolar and the inferior first molar is indicated at a length of 4 to 6 mm from the anatomical tip of the roots, which would completely remove the apical delta of these two teeth. A combination of apical resection and conservative root canal therapy of these two teeth in dogs will produce the best treatment results. With a conservative approach, the safety of root canal healing cannot be guaranteed.

## Figures and Tables

**Figure 1 animals-14-01138-f001:**
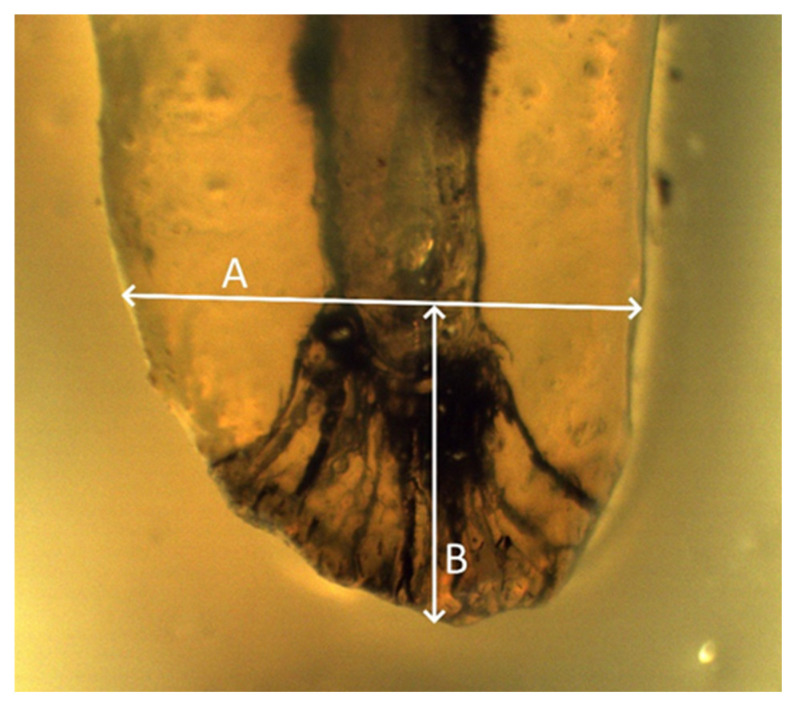
A—root width at the point of initiation of apical ramification; B—length of the apical delta.

**Figure 2 animals-14-01138-f002:**
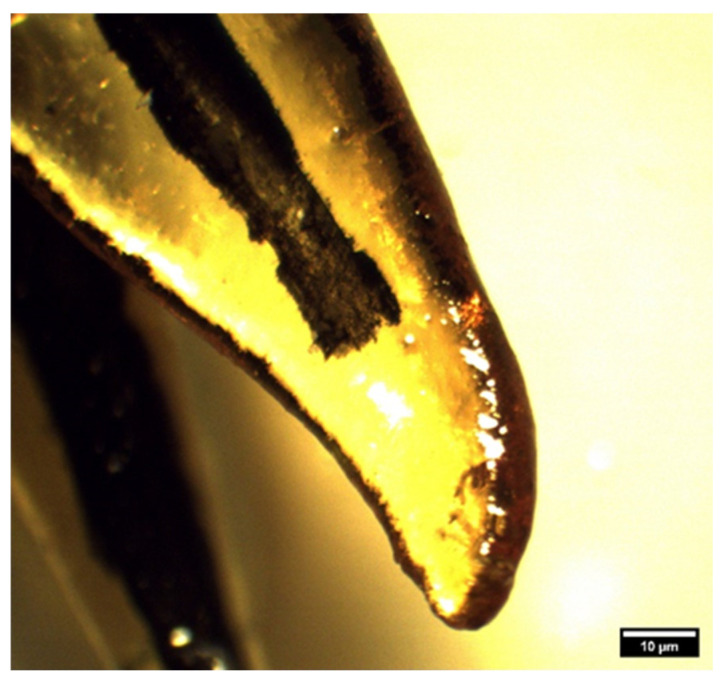
Obliteration of the mesiopalatal root of a superior fourth premolar in a mixed-breed dog (clearing technique, optical magnifier, magnification 10×).

**Figure 3 animals-14-01138-f003:**
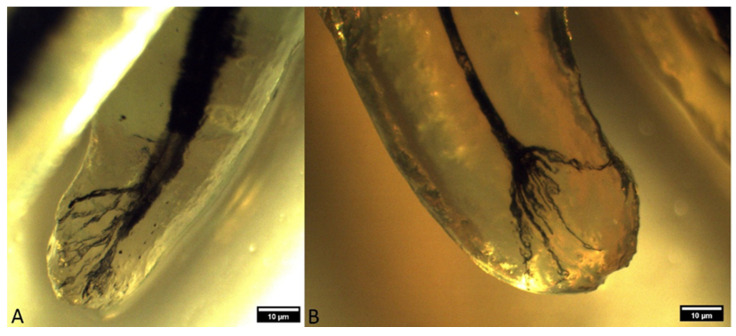
Type II apical delta in superior fourth premolars ((**A**)—mesiopalatal root of a superior fourth premolar in a mixed-breed dog; (**B**)—mesiopalatal root of a superior fourth premolar in a German shepherd) (clearing technique, optical magnifier, magnification 10×).

**Figure 4 animals-14-01138-f004:**
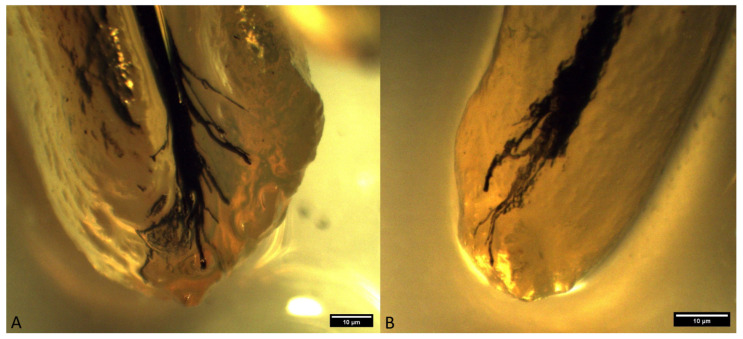
Type II apical delta in inferior first molars ((**A**)—mesial root of an inferior first molar in a mixed-breed dog; (**B**)—mesial root of an inferior first molar in a German shepherd) (clearing technique, optical magnifier, magnification 10×).

**Figure 5 animals-14-01138-f005:**
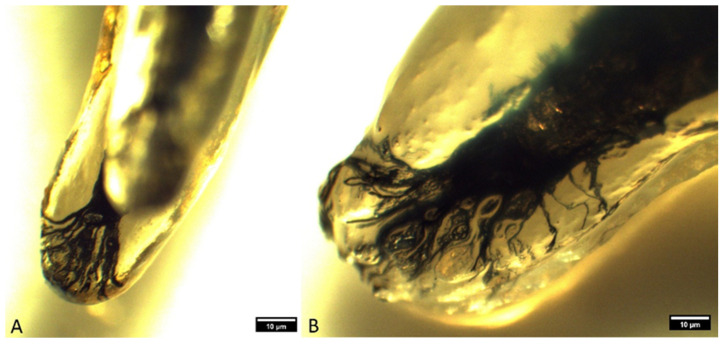
Type IIIA apical delta in superior fourth premolars ((**A**)—mesiopalatal root of a superior fourth premolar in a mixed-breed dog; (**B**)—distal root of a superior fourth premolar in a German shepherd (clearing technique, optical magnifier, magnification 10×).

**Figure 6 animals-14-01138-f006:**
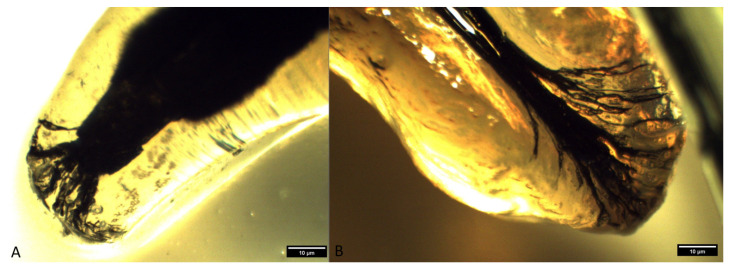
Type IIIA apical delta in inferior first molars ((**A**)—distal root of an inferior first molar in a mixed-breed dog; (**B**)—mesial root of an inferior first molar in a German shepherd) (clearing technique, optical magnifier, magnification 10×).

**Figure 7 animals-14-01138-f007:**
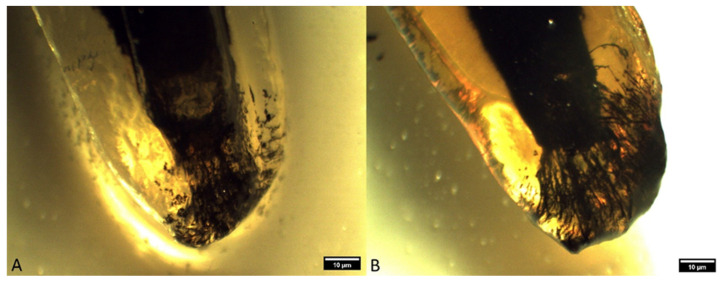
Type IIIB apical delta in superior fourth premolars ((**A**)—distal root of a superior fourth premolar in a German shepherd; (**B**)—distal root of a superior fourth premolar in a mixed-breed dog) (clearing technique, optical magnifier, magnification 10×).

**Figure 8 animals-14-01138-f008:**
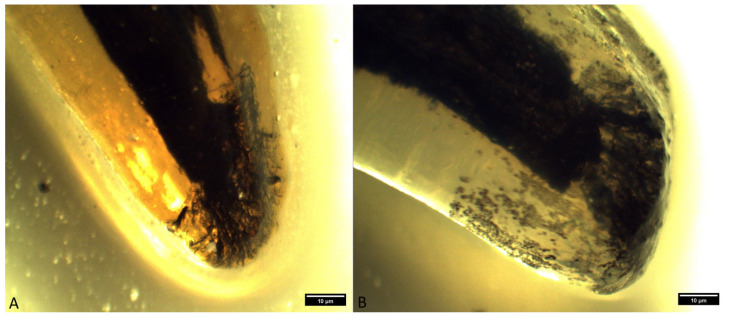
Type IIIB apical delta in inferior first molars ((**A**)—mesial root of an inferior first molar in a mixed-breed dog; (**B**)—distal root of an inferior first molar in a German shepherd) (clearing technique, optical magnifier, magnification 10×).

**Table 1 animals-14-01138-t001:** Root lengths of superior fourth premolars and inferior first molars in mixed-breed dogs and German shepherds (cm) (mb—mesiobuccal, mp—mesiopalatal, dis—distal, mes—mesial).

Breed	Root	Superior Fourth Premolars	Inferior First Molars
		Mean ± SD	Mean ± SD
**Mixed Breed**	mb	11.05 ± 1.59	
	mp	19.37 ± 1.53	
	dis	14.71 ± 1.61	13.32 ± 1.50
	mes		15.79 ± 1.63
**German Shepherds**	mb	11.05 ± 0.36	
	mp	20.03 ± 0.42	
	dis	15.51 ± 0.28	14.64 ± 0.32
	mes		18.19 ± 0.60

**Table 2 animals-14-01138-t002:** Root width at the point of initiation of apical ramification in root canals of right (108) and left (208) superior fourth premolars and right (409) and left (309) inferior first molars in mixed-breed dogs and German shepherds (mm) (mb—mesiobuccal, mp—mesiopalatal, dis—distal, mes—mesial).

Breed	Root		Tooth 108			Tooth 208			Tooth 309			Tooth 409	
**Mixed Breed**		N	Mean	SD	N	Mean	SD	N	Mean	SD	N	Mean	SD
	mb	35	2.61	0.81	34	2.79	0.89						
	mp	32	2.78	0.69	33	2.98	0.73						
	mes							36	4.69	1.55	36	4.66	1.55
	dis	36	3.74	1.13	35	3.87	1.19	36	4.32	1.02	36	4.57	1.02
**German Shepherds**	mb	7	3.13	1.01	7	3.07	1.32						
	mp	7	2.91	0.92	7	3.23	1.09						
	mes							7	4.71	1.97	7	5.07	2.30
	dis	7	4.26	1.47	7	4.42	1.47	7	5.05	1.37	7	5.14	1.22

**Table 3 animals-14-01138-t003:** The length of the apical delta from the site of initiation of apical ramification to the anatomical apex of the roots of the right (108) and left (208) superior fourth premolars and right (409) and left (309) inferior first molars in mixed-breed dogs and German shepherds (mm) (mb—mesiobuccal, mp—mesiopalatal, dis—distal, mes—mesial).

Breed	Root		Tooth 108			Tooth 208			Tooth 309			Tooth 409	
**Mixed Breed**		N	Mean	SD	N	Mean	SD	N	Mean	SD	N	Mean	SD
	mb	35	2.07	0.80	34	2.08	0.95						
	mp	32	2.20	0.83	33	2.37	0.86						
	mes							36	2.66	0.92	36	2.68	1.09
	dis	36	2.53	0.87	35	2.90	1.26	36	2.22	0.68	36	2.45	0.86
**German Shepherds**	mb	7	2.60	0.89	7	2.35	1.02						
	mp	7	2.34	1.37	7	2.52	1.06						
	mes							7	2.97	1.19	7	3.65	1.34
	dis	7	2.80	0.97	7	3.26	1.22	7	2.90	1.13	7	2.84	0.91

## Data Availability

The data presented in this study are available on request from the corresponding author.

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
