# Peer review of "Comparative Morphology of the Carnassial Teeth Root Canals in Mixed-Breed Dogs and German Shepherds"

_animals, 2024, doi:10.3390/ani14081138_

Round 1

Reviewer 1 Report

Comments and Suggestions for Authors

The manuscript is a very interesting research about the teeth canals of the superiorfourth premolar and the Inferior first molar mainly in mixed breed dogs and some German shepherds. The figures are excellent and complement with statistics. It has a good conclusion, wich should be summarized in the abstract. Minor corrections should be performed as below:

Lines:

1-2 : The N is higher in mixed breeds, so it should be before German Shepherds

28-31: Please include a conclusion of this investigation, You have the objective in these lines.

Change the terms maxillary and mandibulary to superior and inferior respectively along the manuscript. Review the Nomina Anatomica Veterinaria 2017

39: Change jaws to mouth

75: Include the methodology to extract the teeth from the maxilla and mandibula.

117 : Cervical?

figures 4, 6 and 8 : include a better visualization of the metric scale just like the other figures

180: 6.06 to 1.73 mm

Author Response

Lines:

1-2 : The N is higher in mixed breeds, so it should be before German Shepherds – corrected

28-31: Please include a conclusion of this investigation, You have the objective in these lines. – included in the abstract

Change the terms maxillary and mandibulary to superior and inferior respectively along the manuscript. Review the Nomina Anatomica Veterinaria 2017 – corrected

39: Change jaws to mouth – corrected

75: Include the methodology to extract the teeth from the maxilla and mandibula. – included in the text

117 : Cervical? – The term 'cervical' is more commonly used in human dentistry, corrected

figures 4, 6 and 8 : include a better visualization of the metric scale just like the other figures – corrected

180: 6.06 to 1.73 mm – corrected

Reviewer 2 Report

Comments and Suggestions for Authors

Dear authors,

I have read your manuscript entitled Comparative Morphology of the Carnassial Teeth Root Canals in German Shepherds and Mixed Breed Dogs with great interest. It is well written. I was especially delighted by your excellent photographs. I have, however, some textual and minor remarks that can be found below in standard typing and a number of major comments that are in bold typing. 

Abstracts

The simple abstract it too abstract and not simple. What is an apical delta? What are the different types? What do they entail? 

Line 15: Knowledge of the morphology …

Keywords: Add 2 more keywords. The current keywords are not specific enough.

Introduction

Line 36-37: Which teeth are known as the carnassials? This sentence is ambiguous. 

Line 48: The final part of the root canal… Describe using anatomical terminology.

Line 50: The main root canal… Of which tooth?

Line 67: Why are specifically German Shepherd dogs compared with mixed breed dogs?

Materials and Methods

General remark: There were much more mongrels than German Shepherds (only 8). Is this small number of 8 compared to 35 enough to perform a comparative morphological study between German Shepherds and mixed-breed dogs? What about the gender and body weight of the dogs? Doesn’t that influence the measured values of the teeth (length, width etc. of the roots for example)

Line 81: Where did you drill? In the root of the tooth, in the crown, …? From the lingual or the buccal/palatal side, …?

Line 92: become > became, moving > bending

Line 95: indicated that the end of the decalcification process was reached.

Lines 113-114: the maxillary fourth premolar and mandibular first molar in which the root canals may bifurcate. They may bifurcate, but not always? Such bifurcation is only present in the carnassials?

Line 117: in the cervical and middle third of the roots. What is the cervical third of the root?

Results:

General remarks regarding the values: What is the value of the absolute values? Should you not give relative values in relation to the length and width of the tooth? The values for the German Shepherds also show a wide variation. Such variation could be expected for the mongrels, but not for the German Shepherds since a smaller variation in body weight is expected here. Or not? What was the body weight of the dogs – German Shepherds and mongrels? I miss a systematic, mathematical approach.

Line 195: Tooth 309

Lines 209-210: mesi-obuccal > mesio-buccal (idem lines 225-226)

Discussion:

General remark: Use paragraphs.

Line 275: The value was…

Line 293: Gamm et al. in their study state …

How does your manuscript differ from that of Hernandez? What does it add? Where is the novelty of your research? Why have you compared German Shepherds with mongrels? What was your hypothesis? Did you perform statistics?

Conclusion: This paragraph does not present the conclusion of your study. It rather fits within the discussion. Please provide a take home message in the conclusion. What should be remember from your study? What have you found that is new?

Author Response

The simple abstract it too abstract and not simple. What is an apical delta? What are the different types? What do they entail? – corrected 

Line 15: Knowledge of the morphology … - corrected

Keywords: Add 2 more keywords. The current keywords are not specific enough. – added

Line 36-37: Which teeth are known as the carnassials? This sentence is ambiguous. - corrected

Line 48: The final part of the root canal… Describe using anatomical terminology. - corrected

Line 67: Why are specifically German Shepherd dogs compared with mixed breed dogs? - Both German Shepherds and mixed breed dogs in our study are mesocephalic dog breeds which is why they were compared.

General remark: There were much more mongrels than German Shepherds (only 8). Is this small number of 8 compared to 35 enough to perform a comparative morphological study between German Shepherds and mixed-breed dogs? What about the gender and body weight of the dogs? Doesn’t that influence the measured values of the teeth (length, width etc. of the roots for example) -

We consider that the number of samples is adequate, since we compared roots, of which 80 belonged to German Shepherds. For our research we used mesocephalic mixed dog breeds whose average length of the dental arch was 9-11 cm, which is very similar to the average length of the dental arch in German Shepherds.

Line 81: Where did you drill? In the root of the tooth, in the crown, …? From the lingual or the buccal/palatal side, …? – explained in the text

Line 92: become > became, moving > bending – corrected

Line 95: indicated that the end of the decalcification process was reached. – corrected

Lines 113-114: the maxillary fourth premolar and mandibular first molar in which the root canals may bifurcate. They may bifurcate, but not always? Such bifurcation is only present in the carnassials? - Thank you for this remark. The subject of our research were carnassial teeth, so we are not aware of the presence of such bifurcation in other teeth. However, this leaves room for future research.

Line 117: in the cervical and middle third of the roots. What is the cervical third of the root? - The term 'cervical' is more commonly used in human dentistry; corrected

General remarks regarding the values: What is the value of the absolute values? Should you not give relative values in relation to the length and width of the tooth? The values for the German Shepherds also show a wide variation. Such variation could be expected for the mongrels, but not for the German Shepherds since a smaller variation in body weight is expected here. Or not? What was the body weight of the dogs – German Shepherds and mongrels? I miss a systematic, mathematical approach.

Thank you for this remark, it is very important to have this morphometric data about roots. We included the average lengths of tooth roots for both superior fourth premolar and inferior first molar in German Shepherds and mixed breed dogs, please find these values in Table 1. We agree with your statement about wide variation, however these are the results we measured. Regarding the body weight of the dogs, we didn't have that data, however, considering that both mixed breed dogs and German Shepherds are mesocephalic breeds with similar length of the dental arch, we considered this comparison adequate for the research.

Line 195: Tooth 309 - corrected

Lines 209-210: mesi-obuccal > mesio-buccal (idem lines 225-226) – corrected

General remark: Use paragraphs.

Line 275: The value was… - corrected

Line 293: Gamm et al. in their study state … - corrected

How does your manuscript differ from that of Hernandez? What does it add? Where is the novelty of your research? Why have you compared German Shepherds with mongrels? What was your hypothesis? Did you perform statistics? -

The research conducted by Hernandez et al. was the basis of our research which differs in the parameters that were determined, i.e. root width at the point of initiation of apical ramification. Breed and age of the dogs was also determined, which is not the case in the study conducted by Hernandez et al. Additionally, the classification according to Watanabe et al. was used in our research. The significance of this study is that it offers data on the frequency, length and width of apical deltas in carnassial teeth using representative pictures. As mentioned above, both German Shepherds and mixed breed dogs in our study are mesocephalic dog breeds which is why they were compared. Our hypothesis was whether there is a difference between these two types of dogs. Only descriptive statistics were used.

Conclusion: This paragraph does not present the conclusion of your study. It rather fits within the discussion. Please provide a take home message in the conclusion. What should be remember from your study? What have you found that is new? – corrected

Reviewer 3 Report

Comments and Suggestions for Authors

The paper includes a very detailed and meticulously conducted study on the morphology of the apical portion of the roots in the carnassial teeth (P4/m1) in dogs. The study presents a substantial sample of the studied teeth, almost 250 specimens (collected from euthanized dogs).While presenting the apical root morphology in great details, the study is very narrowly cut. In fact, it is restricted to the apical root morphology in German Shepherds and some mixed-breed dogs. My first question to the authors: What is the basis on this classification? For example, were these German Shepherds purebreds with pedigree, or just checked against a dog type?

The authors mention the endodontic procedures which can be performed on the dogs and that the knowledge on the apical morphology is important for such treatment but they did not offer any greater details of the said endodontic treatment. The study apparently lacks any in vivo method of detection of the apical root morphology; thus, it seems of a limited value in terms of dental diagnostics. It allows only to 'predict' the apical morphology based on given percentages, observed in the studied specimens (postmortem). Maybe Authors could add some information to broaden the scope of the study and elaborate a bit more on the implementation or clinical meaning of the presented results. Anyway, it is a well-presented and thoroughly researched study.

A minor remark: in ref. [23] should be ‘scanning’ in the title.

Author Response

The paper includes a very detailed and meticulously conducted study on the morphology of the apical portion of the roots in the carnassial teeth (P4/m1) in dogs. The study presents a substantial sample of the studied teeth, almost 250 specimens (collected from euthanized dogs).While presenting the apical root morphology in great details, the study is very narrowly cut. In fact, it is restricted to the apical root morphology in German Shepherds and some mixed-breed dogs. My first question to the authors: What is the basis on this classification? For example, were these German Shepherds purebreds with pedigree, or just checked against a dog type?

As mentioned above, both German Shepherds and mixed breed dogs in our study are mesocephalic dog breeds which is why they were compared. The German Shepherds in this study were not purebreds with pedigree.

The authors mention the endodontic procedures which can be performed on the dogs and that the knowledge on the apical morphology is important for such treatment but they did not offer any greater details of the said endodontic treatment. The study apparently lacks any in vivo method of detection of the apical root morphology; thus, it seems of a limited value in terms of dental diagnostics. It allows only to 'predict' the apical morphology based on given percentages, observed in the studied specimens (postmortem). Maybe Authors could add some information to broaden the scope of the study and elaborate a bit more on the implementation or clinical meaning of the presented results. Anyway, it is a well-presented and thoroughly researched study.

This study focuses on the apical delta, a type of root ramification that has important therapeutic implications. Because of the limited resolution of radiographs in vivo, apical deltas cannot be observed. As a result, an in vitro study like this one is the most effective strategy to alert veterinary dentistry therapists to its existence. Due to the prevalence of apical delta, endodontic therapy alone is insufficient, and endodontic surgery must be considered as part of the treatment.

A minor remark: in ref. [23] should be ‘scanning’ in the title. - corrected

Round 2

Reviewer 2 Report

Comments and Suggestions for Authors

Dear authors,

Thank you for revising the manuscript according to my suggestions. I have no further remarks.